# Lattice Thermal Conductivity of Monolayer InSe Calculated by Machine Learning Potential

**DOI:** 10.3390/nano13091576

**Published:** 2023-05-08

**Authors:** Jinsen Han, Qiyu Zeng, Ke Chen, Xiaoxiang Yu, Jiayu Dai

**Affiliations:** 1Department of Physics, National University of Defense Technology, Changsha 410073, China; 2Hunan Key Laboratory of Extreme Matter and Applications, National University of Defense Technology, Changsha 410073, China

**Keywords:** thermal conductivity, deep potential, phonon scattering

## Abstract

The two-dimensional post-transition-metal chalcogenides, particularly indium selenide (InSe), exhibit salient carrier transport properties and evince extensive interest for broad applications. A comprehensive understanding of thermal transport is indispensable for thermal management. However, theoretical predictions on thermal transport in the InSe system are found in disagreement with experimental measurements. In this work, we utilize both the Green–Kubo approach with deep potential (GK-DP), together with the phonon Boltzmann transport equation with density functional theory (BTE-DFT) to investigate the thermal conductivity (κ) of InSe monolayer. The κ calculated by GK-DP is 9.52 W/mK at 300 K, which is in good agreement with the experimental value, while the κ predicted by BTE-DFT is 13.08 W/mK. After analyzing the scattering phase space and cumulative κ by mode-decomposed method, we found that, due to the large energy gap between lower and upper optical branches, the exclusion of four-phonon scattering in BTE-DFT underestimates the scattering phase space of lower optical branches due to large group velocities, and thus would overestimate their contribution to κ. The temperature dependence of κ calculated by GK-DP also demonstrates the effect of higher-order phonon scattering, especially at high temperatures. Our results emphasize the significant role of four-phonon scattering in InSe monolayer, suggesting that combining molecular dynamics with machine learning potential is an accurate and efficient approach to predict thermal transport.

## 1. Introduction

Two-dimensional (2D) materials enable the field effect transistor (FET) to keep shrinking due to their free dangling bond surface and high carrier mobility [1,2]. Post-transition metal chalcogenides (PTMCs), including InSe, GaSe, GaTe, and GaS, are a family of van der Waals 2D materials that are distinguished in 2D FET applications and optoelectronics [3]. The few-layered PTMCs have a wide range of band gap [4,5], exhibit large, tunable photoluminescence [6], and have high carrier mobility [7,8], thanks to their weak van der Waals interaction and dielectric permittivity. This makes them promising candidates for ultrathin and flexible FET devices [9], photovoltaic [10], and photodetector [11,12]. Understanding the thermal transport in 2D PTMCs is crucial for the thermal management of PTMCs-based FET devices and optoelectronics.

Indium selenide (InSe), as a typical member of PTMCs, has drawn wide attention and has been synthesized experimentally [13]. There have been some experimental measurements and theoretical calculations focusing on the thermal conductivity κ of InSe. Through a time-domain thermoreflectance approach, Rai et al. measured the in-plane κ of mechanically exfoliated few-layer InSe on Si substrate, obtained a value of 8.5 ± 2 W/mK, and showed the thickness-independent in-plane κ when the thickness is larger than 100 nm [14]. By using power-dependent Raman spectroscopy and an approximate radial heating flow model, Li et al. reported the in-plane thermal conductivity as large as about 53.1 and 38.2 W/mK for thin-layer InSe with and without the Al_2_O_3_ capping layer, respectively [15]. Buckley et al. performed scanning thermal microscopy measurements of few-layer InSe and observed an enhanced heat dissipation with the increasing sample size from about 0.33–2.98 μm [16]. After careful scrutiny, we favor the results of Rai et al.’s work as a reference in the following discussions.

Untill now, theoretical predictions of κ of InSe monolayer give much larger values than Rai et al.’s results and vary widely from 27.60 to 63.73 W/mK [17,18,19,20,21,22]. Given the fact that the previous phonon Boltzmann transport equation with density functional theory (BTE-DFT) calculations only consider three-phonon scatterings, the origin of the overestimation of κ might arise from the exclusion of high-order phonon scatterings. The significance of four-phonon scattering has been proved in calculating κ of a broad range of materials, such as cubic BAs [23,24,25,26], monolayer graphene [27], and SnSe [28]. However, the effects of four-phonon scattering and even higher-order phonon scatterings on thermal transport in InSe remains to be clarified.

Molecular dynamics (MD) can capture the full-order phonon scattering processes, and therefore can predict the κ more accurately. However, ab initio MD is highly time-consuming for the calculation of κ. Non-equilibrium MD approach requires a steady temperature gradient after a long-time energy relaxation process, while the equilibrium MD approach requires a large correlation time for the convergence of integral based on the Green–Kubo (GK) formula [29,30]. Classical MD is an efficient approach to investigating thermal transport. Pham et al. employed Stillinger–Weber (SW) potential to conduct non-equilibrium MD simulations and reported a much larger κ than Rai et al.’s experimental value [31]. An accurate interatomic potential model is the key to classical MD simulations. The emerging machine learning potential, such as deep potential (DP) and momentum tensor potential (MTP), can describe the interatomic interactions with DFT accuracy and acceptable computational cost [32,33,34,35]. These machine learning approaches have been successfully implemented in studies on thermal transport [36,37,38,39,40,41].

In this work, we perform a neural network DP model based MD simulations to predict the κ of monolayer InSe based on the basis of GK formula. The GK-DP results are in good agreement with Rai et al.’s experimental results. We also calculate the κ by the BTE-DFT approach by considering only three-phonon scattering scenario. We find that the BTE-DFT approach overestimates the κ, which is unsurprising in the shield of with previous BTE calculations. Whilst through the analysis of phase space of four-phonon scattering and mode-decomposed contribution on κ, the significant role of four-phonon scattering in the thermal conductivity calculation has been proved, due to the large gap between acoustic and upper optical branches.

## 2. Methodology

The structure of monolayer InSe monolayer exhibits a quadruple layer in the Se-In-In-Se sequence, as shown in Figure 1a,b. It shows a P$\bar{6}$m2 space group symmetry with buckled surface structures along *z*-direction as well as a honeycomb structure in xy-plane. Based on our first-principles calculations, the optimized lattice constant is 4.087 Å, the In-In bond and In-Se bond are 2.825 and 2.682 Å respectively, and the layer thickness is 5.373 Å, which are in good agreement with previous studies [42,43,44]. The effective thickness of monolayer InSe used to calculate the κ is 8.57 Å, which contains the interlayer distance of the AA-stacked InSe.

The DP model for InSe is generated with DeePMD-kit packages [45] and VASP packages [46,47]. A concurrent learning scheme, Deep Potential Generator (DP-GEN) [48] has been adopted to sample an adequate data set. We consider 2×2×1 supercells (36 atoms) and 3×3×1 supercells (64 atoms) as the initial configurations and run molecular dynamics under NVT ensembles, where the temperature ranges from 200 K to 600 K. The training sets consist of 610 configurations for 36-atom-system and 360 configurations for 64-atom-system. In the DFT labeling procedure, The PBE exchange-correlation functional is used [49], and the pseudopotential takes the projector augmented-wave (PAW) formalism [50,51]. the kinetic energy cutoff is set to 600 eV and the spacing between k points in the Brillouin zone is chosen as 0.25 ${\overset{˚}{A}}^{-1}$ to guarantee the accuracy of energies and forces.

For DP training, the embedding network is composed of three layers (25, 50, and 100 nodes) while the fitting network has three hidden layers with 240 nodes in each layer. The total number of training steps is set to 1,000,000. The cutoff radius $r_{c}$ is chosen to be $6.0\;\overset{˚}{A}$. The weight parameters in loss function for energies $p_{e}$, forces $p_{f}$, and virials $p_{V}$ are set to (0.02,1000,0.02) at the beginning of training and gradually change to (1.0,1.0,1.0). The accuracy of the DP model is tested in the whole sampling configuration. As presented in Figure 2, the root mean square error (RMSE) of energy is 0.337meV/atom and the RMSE of forces is $0.101\;\mathrm{eV}/\overset{˚}{A}$. As can be seen, the scatters of potential energy and atomic force locate around the dashed diagonal, indicating the high accuracy of the trained DP model.

In DPMD simulations, the κ is obtained from the integration of the heat current autocorrelation function (HCACF), known as Green–Kubo equation [52,53]
$$\kappa_{\alpha \beta }=\frac{V}{k_{B}T_{i}^{2}}\int_{0}^{\infty }\langle J_{\alpha }\left(0\right)J_{\beta }\left(t_{\mathrm{corre}}\right)\rangle dt_{\mathrm{corre}}$$
where $t_{\mathrm{corre}}$ is the heat current autocorrelation time, and $J_{\alpha }$ is the α-th component of the full heat current vector J, which is typically computed as [54],
$$J=\frac{1}{V}\left(\sum_{i}v_{i}\epsilon_{i}+\sum_{i}\Xi_{i}\cdot v_{i}\right)$$

Here, $\epsilon_{i}$ and $\Xi_{i}$ are the energy and stress tensor of atom *i*.

We perform all the DPMD simulations with the LAMMPS package [54]. The time step is set to 1 fs and the Nosé-Hoover thermostat [55,56] is employed in the NVT ensemble. After a thermalization stage of 20 ps, the ensemble is switched into the NVE ensemble to calculate the heat current autocorrelation function during the next 1.6 ns. Each simulation is run 15 times, which had independent initial velocity distributions, to provide a representative sample for the relevant statistical analysis. As shown in Figure 3a and the inset, the normalized HCACF decays within around 200 ps, and finally fluctuates around zero after 200 ps. After that, the κ can be obtained by the integral of HCACF. Figure 3b demonstrates that the averaged integral of HCACF increases and gradually converges as the time of integral upper limit increases. As can be seen, an integral upper limit of 200 ps is large enough to obtain a converged κ. The inset shows the size dependence of κ. A simulation cell size of 20×20×1 (1600 atoms), corresponding to a side length of about 8 nm, is large enough to overcome the size effect.

The density-functional theory (DFT) [57] is carried out by the Quantum-ESPRESSO package [58,59]. The Optimized Norm-Conserving Vanderbilt (ONCV) pseudopotential [60] is adopted to describe the core electronic behaviors. The Perdew–Burke–Ernzerhof (PBE) exchange-correlation [49] functional is implemented in DFT calculations. A plane-wave basis setup with a kinetic energy cutoff of 90 Ry for the wavefunction is used. 30×30×1 Monkhorst-Pack grid [61] with fixed electron distribution are implemented. At least 25 Å vacuum space with truncated Coulomb potential [62] is implemented to avoid spurious interactions induced by the periodic boundary condition. The spin-orbit coupling effects is considered in the band structure calculations. In the structures relaxation process, the Hellman-Feynman forces converged lower than $10^{-10}$ Ry/Bohr. A 6×6×1 supercell (a size of 24.522 Å) with 2×2×1 k grid and 5×5×1 (a size of 20.435 Å) supercell with 2×2×1 k grid is used to calculate the second- and third-order interatomic force constants (IFCs), respectively. Specifically, a cutoff of the 8th nearest neighbor for the interaction range is taken into consideration to obtain the third-order IFC. The Phonopy package [63] is used to generate structures in second-order IFCs calculation and ShengBTE [64] package is adopted in third-order IFCs and BTE calculations. For comparison with experiment measurement, the isotope scattering, Born effective charges and dielectric constants are considered in the BTE calculations.

In the BTE framework, the temperature-dependent thermal conductivity can be described by
(1)$$\kappa_{\alpha \beta }=\frac{1}{k_{B}T_{i}^{2}NV}\sum_{\lambda }{\left(\hbar \omega_{\lambda }\right)}^{2}f_{\lambda }\left(1+f_{\lambda }\right)v_{\lambda }^{\alpha }F_{\lambda }^{\beta }$$
where $\kappa_{\alpha \beta }$ is the αβ direction component of the thermal conductivity tensor, *V* is the volume of the material system, $k_{B}$ is the Boltzmann constant, and *T* is the temperature. λ indicates the phonon mode and *f* denotes the Bose-Einstein distribution. *v* is the phonon group velocity, and *F* is the projection of mean free displacement.

Due to the very high computational cost of four-phonon scattering rates, only three-phonon scattering is taken into consideration in our BTE calculations. The thermal conductivity κ is calculated by the iterative solution of BTE. After a careful parameter test for the convergence of BTE calculations, as shown in Figure 4, the q-grid is set as 100×100×1, and the value of scalebroad factor is set as 1.5.

## 3. Results and Discussion

The electronic structures calculated by DFT are shown in Figure 5a. There is a narrowly-distributed “Mexican hat” shape band structure around the top of the first valance band, which indicates that the electronic density of states can form a van Hove singularity. The spin-orbit coupling is considered in the calculation of band structures. There is no splitting around the Γ point and a slight splitting at the M and K points. The InSe monolayer is a typical indirect gap semiconductor with the valance band maximum (VBM) located along the K-Γ path while the conduction band minimum (CBM) is located at the Γ point. The direct band gap (defined by the gap at Γ point) is about 1.37 eV. Therefore, the electron-phonon coupling effect on κ is ignored in this work. The thermal conductivity is fully attributed to the lattice vibration.

The electron localization function (ELF) is plotted in Figure 5b. It can be seen that along the In-Se bond, the ELF value is about 0.8 around the Se atom while 0.5 around the In atom, which indicates that the electrons prefer localizing toward Se atom. However, along In-In bond, the ELF has the maximum value at the center of In-In bond, reaching 0.9, which shows that there is no charge transfer between In atoms. Therefore, in strong polar crystals, the long-range electrostatic interactions would have an impact on thermal transport. In our calculations, the Born effective charge and dielectric constants are taken into consideration.

The phonon dispersion curves calculated by DP and DFT are shown in Figure 6a. As can be seen, DP reproduces the phonon dispersion by DFT well, thus validating the accuracy of the trained DP model. Due to the four atoms in the primitive cell of monolayer InSe, there are three acoustic branches and nine optical branches. For the lowest three acoustic branches, the highest phonon frequency is 2 THz at the K point. As can be seen, the lower two optical branches (below 2.6 THz) also exhibit large dispersion and are found to contribute much to κ as will be shown later. An optical branch around 3 THz locates in the middle energy regime. The frequency values of the upper six optical branches are larger than 4.7 THz. There is a large energy gap (about 2.1 THz) between the the upper six optical branches and the lower two optical branches. This large energy gap prohibits a number of three-phonon scattering channels involving two phonons in the lower five branches and one phonon in the upper six optical branches. For example, when phonon frequency is larger than the maximum of the energy sum of lower five branches and middle optical branch, there is a sharp decrease of three-phonon scattering phase space (see details later). Therefore, the optical branch around 3 THz play an important role to bridge the lower and upper branches and participate in a lot of acoustic-optical phonon scatterings. Hence, the phonon dispersion implies the necessity of considering four-phonon scattering, as will be discussed later.

The κ results are shown in Figure 6b. The thickness of InSe monolayer used in this work is 8.57 Å, which includes the interlayer distance. The κ calculated by DP-GK is 9.52 ± 0.86 W/mK at room temperature (300 K), which is in good agreement with the experimental value (8.5 ± 2 W/mK) reported in Ref. [14]. For the convenience of comparison, the κ values in previous works are recalculated for consistently including the interlayer distance, as shown in Table 1. Pham et al.’s MD result is about 46 W/mK with a thickness of 5.385 Å [31], corresponding to about 28.9 W/mK with a thickness of 8.57 Å, which is much larger than DPMD and experimental results.

By using the BTE-DFT approach, the predicted κ in this work is 13.08 W/mK at 300 K, which is close to Majumdar et al.’s BTE-DFT results (28.2 W/mK with a thickness of 5.380 Å, corresponding to 17.7 W/mK with a thickness of 8.57 Å) [20]. Our result is lower than other previous BTE-DFT results, as shown in Table 1. All the κ values by BTE-DFT, which only consider the three-phonon scattering, are larger than the DPMD and experimental results. Therefore, phonon-phonon scatterings beyond third-order anharmonicity, which are missing in current BTE calculations and included in the MD simulations, should be the origin of this discrepancy between BTE-DFT and DPMD results.

To illustrate the significance of four-phonon scattering, we calculate the phase space of three-phonon (P${}_{3}$) and four-phonon (P${}_{4}$) scatterings, as depicted in Figure 7a. Generally, P${}_{3}$ is larger than P${}_{4}$. But it should be noted that there are three frequency regimes where P${}_{4}$ is larger than P${}_{3}$, i.e., near 1.1 THz, 1.6∼2.1 THz, and 5∼5.8 THz. The phonons of lower optical branches near the Γ point contribute to the sharp decrease and subsequent increase of P${}_{3}$ in the first frequency regimes mentioned above (near ∼1.1 THz). These phonons have small momentum and can only participate in the three-phonon scattering processes with small momentum change. Their energy of about 1.1 THz is too large for the most of phonon scattering processes within the lower six branches (three acoustic branches and three lower optical branches), but too small for the phonon scattering across the energy gap between the lower six branches and upper optical branches. In the second frequency regimes (1.6∼2.1 THz), many phonons can participate in the four-phonon scatterings with the splitting, redistribution, and combination of high-frequency phonons in upper optical branches. Phonons in the third frequency regimes (5∼5.8 THz) contribute to the combination process of four-phonon scatterings, thus leading to much larger P${}_{4}$ than P${}_{3}$. Therefore, the four-phonon scattering is not negligible and could lead to a reduction in thermal conductivity.

To evaluate the contribution of different phonons to κ, especially the phonons whose P${}_{4}$ are larger than P${}_{3}$, we further analyze the mode-decomposed phonon properties. Group velocity is a key quantity to evaluate the κ. Figure 7b shows the group velocities of phonons in the second frequency regimes of 1.6∼2.1 THz are close to those of acoustic branches. Noting that the lifetimes of these phonons are as long as about 10 ps, as illustrated in Figure 7c, these phonons are belived to have a certain contribution to total κ. Except for the highest optical branch, the phonons in the upper optical branches in the third frequency regimes (5∼5.8 THz) exhibit anomalous large lifetimes, which stem from the underestimated three-phonon scattering channels prohibited by the large frequency gap. However, when considering four-phonon scattering, the large four-phonon scattering phase space would reduce the lifetimes of phonons in the second and third frequency regimes.

We calculate the cumulative κ as a function of frequency based on BTE-DFT calculations. Here, due to the very high computational cost of four-phonon scattering rates [66], only three-phonon scatterings are included. As shown in Figure 7d, phonons below 2.6 THz (as denoted by the first dashed line), namely phonons in three acoustic branches and lower two optical branches, contribute to the most of κ. As is known, the inclusion of four-phonon scattering would reduce thermal conductivity [67]. Especially for phonons in the second frequency regimes mentioned above (1.6∼2.1 THz), given the fact that these phonons have a certain contribution to total κ, their larger P${}_{4}$ than P${}_{3}$ would further reduce their contribution to κ if considering four-phonon scattering. Besides, the phonons in the upper optical branches except for the highest branch also have a few contributions to total κ, due to the overestimation of their lifetimes when only considering three-phonon scattering. Therefore, the smaller κ calculated by GK-DP than that calculated by BTE-DFT could be attributed to the missing four-phonon scattering in BTE calculations.

Besides, the negative temperature dependence of κ is shown in Figure 6b. When we only consider the three-phonon scattering, the decrease of κ calculated by BTE-DFT follows the 1/T dependence, because of the 1/T dependence of phonon lifetime as shown in Figure 7c. The difference between κ calculated by GK-DP and BTE-DFT increases with increasing temperature, due to the stronger high-order phonon scattering at higher temperatures. The dominant contribution from phonons below 2.6 THz remains unchanged as temperature increases. The cumulative κ as a function of mean free path (MFP) at different temperatures is presented in Figure 8. The maximum MFP reaches as large as about 800 nm, and decreases with increasing temperature, becoming smaller than 500 nm at 500 K. Besides, the large MFP suggests that reducing the sample size is an effective way to reduce the κ and enhance its thermoelectric performance.

## 4. Conclusions

We investigated the lattice thermal conductivity κ of monolayer InSe. Based on the DP model, a machine learning interatomic potential with DFT accuracy was generated and then used to predict the κ as 9.52 W/mK at 300 K, which is found to be close to the experimental results. Meanwhile, we utilized the BTE-DFT to calculate the κ and obtained a larger value (13.08 W/mK at 300 K) than the experimental counterpart, since all the current BTE method only takes three-phonon scattering into consideration and neglect the higher-order phonon-phonon scattering process. The phase space analysis also suggests that the four-phonon scattering is not negligible for low-frequency phonons with large group velocities. Moreover, the upper optical phonons exhibit anomalous large lifetimes. This overestimation originates from the underestimated scattering channels prohibited by the large frequency gap between the lower and upper branches when excluding four-phonon scatterings. The cumulative κ underlines the necessity of considering four-phonon scattering in monolayer InSe because of the certain contribution of these phonons with overestimated lifetimes. We found that the difference between GK-DP and BTE-DFT results enlarges along with increasing temperature, due to the increasing influences from high-order phonon-phonon scattering at high temperatures. Our results suggest that combining molecular dynamics with machine learning potential is an accurate and efficient approach to predicting thermal transport, and also address the significant effect of four-phonon scattering on thermal transport in monolayer InSe.

## Figures and Tables

**Figure 1 nanomaterials-13-01576-f001:**
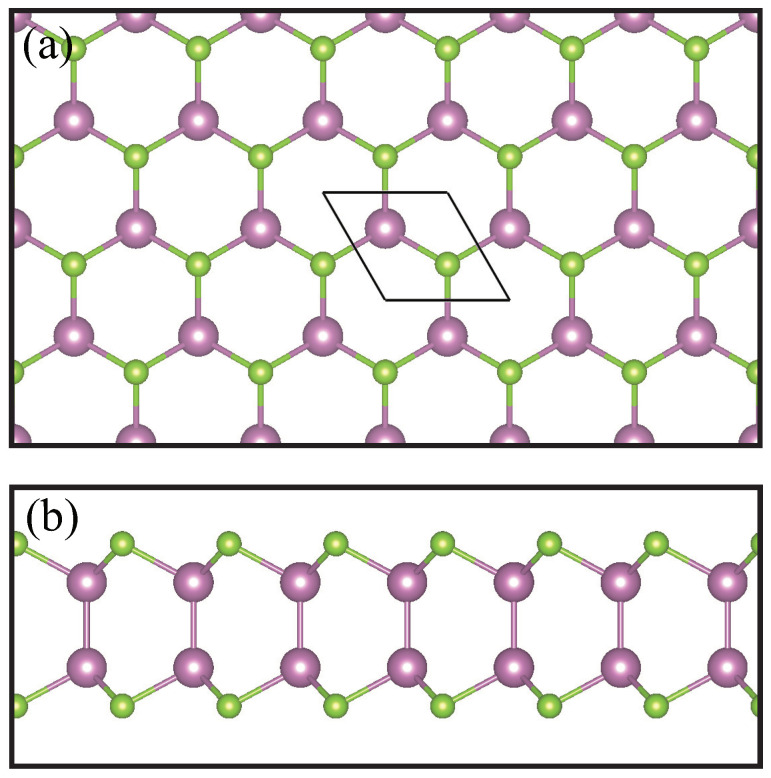
(Color online) Atomic structure of monolayer InSe. (**a**) is the top view and (**b**) is the side view. The In and Se atoms are in purple (large size) and green (small size), respectively. The diamond box in (**a**) denotes the primitive cell.

**Figure 2 nanomaterials-13-01576-f002:**
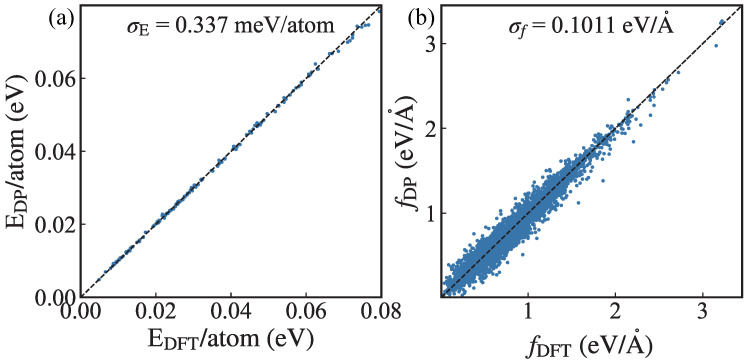
(Color online) (**a**) Potential energies (E) and (**b**) atomic forces (f) from MD simulations using DP and ab initio calculations. The root mean square error (RMSE) of energy $\sigma_{E}$ is 0.337meV/atom and the RMSE of forces $\sigma_{f}$ is $0.101\;\mathrm{eV}/\overset{˚}{A}$. The dashed line is a guide to evaluate the accuracy of the DP model. As can be seen, the scatters of potential energy and atomic force locate near the dashed line, indicating the high accuracy of the trained DP model.

**Figure 3 nanomaterials-13-01576-f003:**
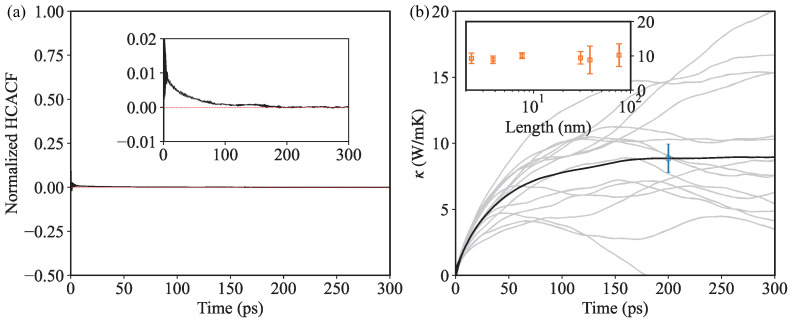
(Color online) (**a**) Normalized heat current autocorrelation function (HCACF) at 300 K as a function of correlation time. The inset in (**a**) shows the enlarged view. (**b**) Thermal conductivity κ calculated by DPMD at 300 K. The inset in (**b**) presents the size dependence of κ.

**Figure 4 nanomaterials-13-01576-f004:**
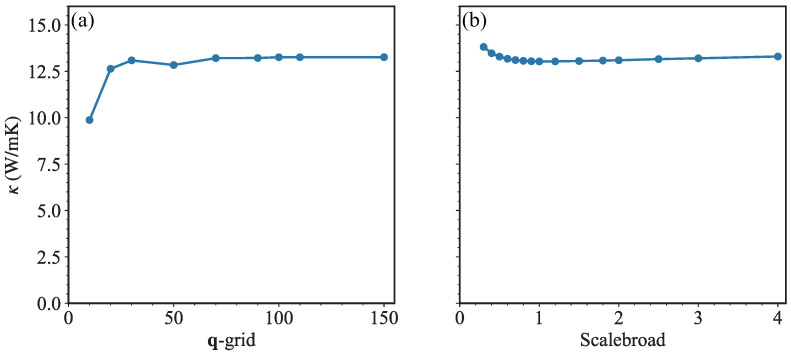
(Color online) Thermal conductivity κ at 300 K as a function of (**a**) q-grid and (**b**) scalebroad in BTE calculations.

**Figure 5 nanomaterials-13-01576-f005:**
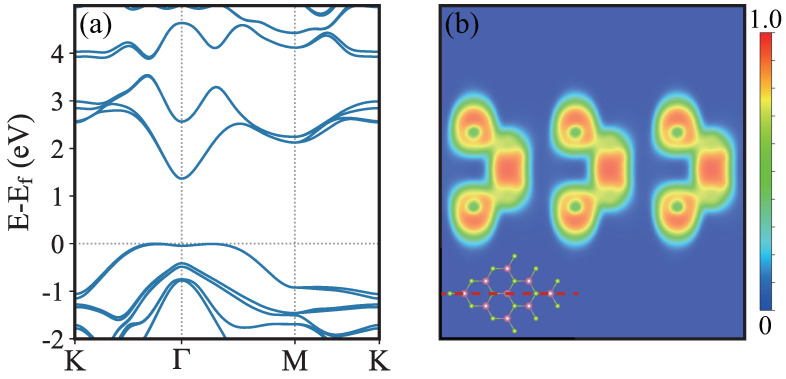
(Color online) Electronic structure of monolayer InSe. (**a**) is the band structure, where the valance band maximum is shifted to 0 eV. (**b**) is the electron localization function (ELF), with the color from blue to red denotes 0 to 1. The inset in (**b**) illustrates the slice plane (red dashed line) for the ELF calculation.

**Figure 6 nanomaterials-13-01576-f006:**
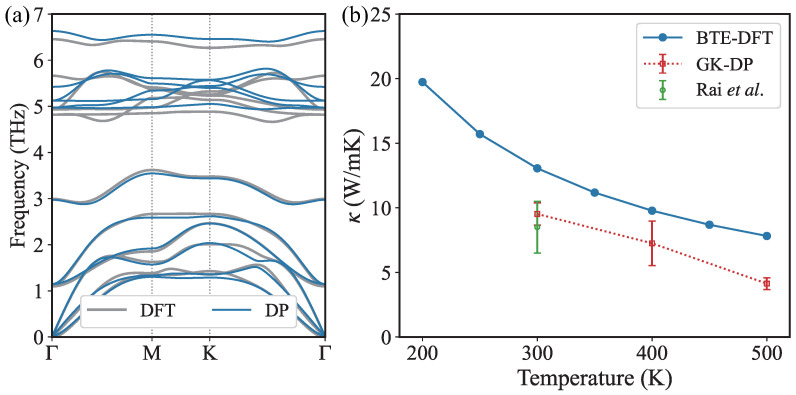
(Color online) (**a**) Phonon dispersion for the InSe monolayer. The gray lines are calculated by DFT while the blue lines are obtained by DP. (**b**) Thermal conductivity κ calculated by BTE-DFT (blue solid line) and GK-DP (red dashed line with error bar). The green dot with an error bar is an experimental value reported in Ref. [14]. The GK-DP results are in good agreement with the experimental measurements but lower than the BTE-DFT results.

**Figure 7 nanomaterials-13-01576-f007:**
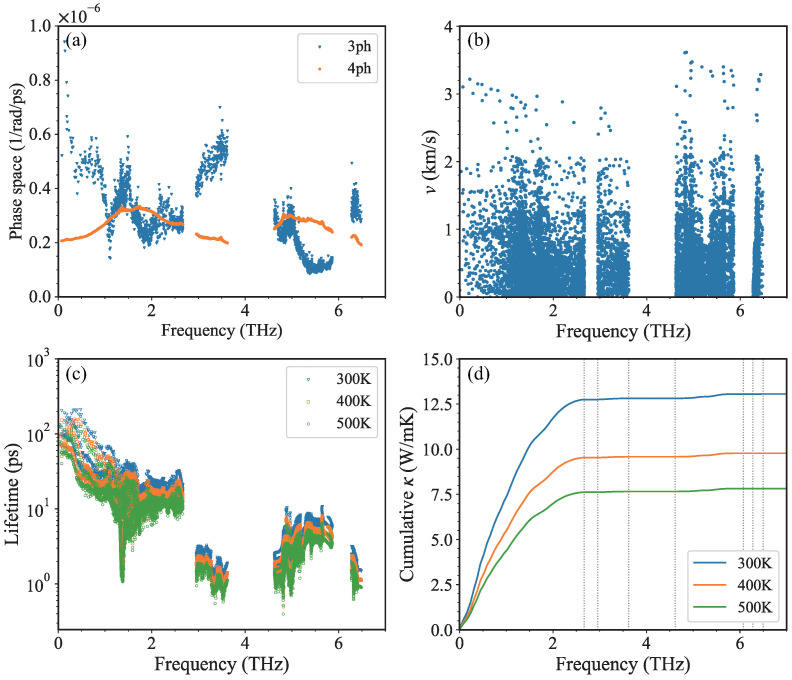
(Color online) (**a**) Phase space of phonon scattering as a function of phonon frequency. The blue triangles represent the phase space for three-phonon scattering events (P${}_{3}$), and the orange circles represent the phase space for four-phonon scattering events (P${}_{4}$). (**b**) Group velocity as a function of phonon frequency. (**c**) Phonon lifetime as a function of phonon frequency at different temperatures. (**d**) Cumulative κ as a function of phonon frequency at different temperatures. In (**c**,**d**), the results at 300 K, 400 K, and 500 K are labeled in blue, orange, and green, respectively. The vertical dashed lines in (**d**) denote the edges of the phonon band gap.

**Figure 8 nanomaterials-13-01576-f008:**
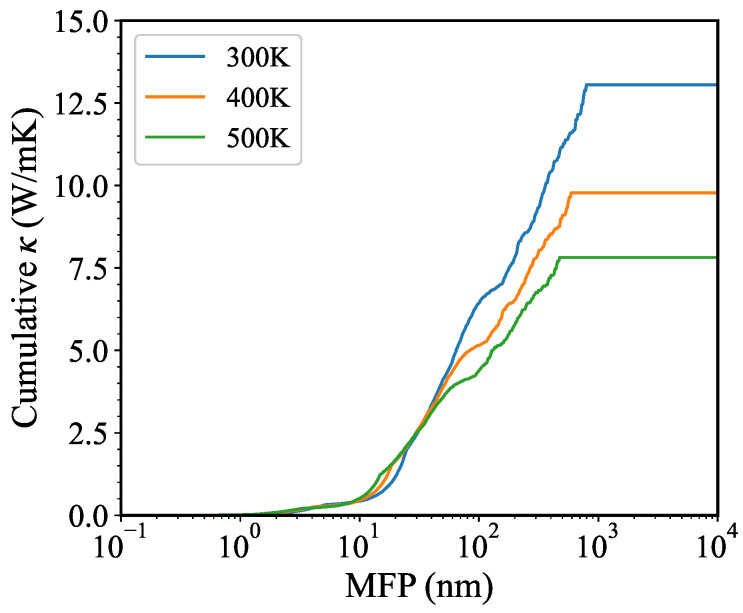
(Color online) Cumulative κ of monolayer InSe as a function of mean free path (MFP) at different temperatures. The results at 300 K, 400 K, and 500 K are labeled in blue, orange, and green, respectively.

**Table 1 nanomaterials-13-01576-t001:** The κ of monolayer InSe at 300 K. The values are recalculated for consistently using the same thickness (d) of 8.57 Å, which includes the interlayer distance.

Method	κ (W/mK)	Thickness (Å)	Recalculated κ (W/mK)	Thickness (Å)	Ref.
Exp.	8.5	-	-	-	Ref. [14]
DP-GK	9.52	8.57	-	-	This work
SW-GK	∼46	5.385	28.9	8.57	Ref. [31]
BTE	13.08	8.57	-	-	This work
BTE	28.20	5.380	17.7	8.57	Ref. [20]
BTE	27.60	8.32	26.8	8.57	Ref. [19]
BTE	41.46	5.381	26.0	8.57	Ref. [17]
BTE	44.30	5.386	27.8	8.57	Ref. [18]
BTE	41.60	5.386	26.1	8.57	Ref. [65]
BTE	63.73	5.381	40.0	8.57	Ref. [21]

## Data Availability

The database can be found at https://github.com/mingzhong15/InSe (accessed on 4 May 2023).

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
