# Peer review of "Lattice Thermal Conductivity of Monolayer InSe Calculated by Machine Learning Potential"

_nanomaterials, 2023, doi:10.3390/nano13091576_

Round 1

Reviewer 1 Report

In this work, authors employed first-principles density functional theory (DFT) calculations within the PBE/GGA method+phonon Boltzmann transport equation to investigate the thermal conductivity of monolayer InSe. They also employ machine learning interatomic potential and Green-Kubo approach to study thermal conductivity. The study is well-conducted, and I can thus recommend this manuscript for publication, after the authors address the following concerns:
1- Were the isotope scattering, Born effective charges and dielectric constants considered in the BTE solution?
2-The results for the convergence of the thermal conductivity with respect to the Q-grid has to be presented (though 90*90*1 should be normally fine)
3- As we know, long-range electrostatic interactions in strong polar crystals will lead to the famous LO (longitudinal optical)-TO (transverse optical) splitting and finally impact on thermal conductive properties more or less. Please plot electron localization function (ELF) to more clearly investigate the bonding nature in these systems and confirm the systems are covalent, otherwise Born effective charges and dielectric constants may need to be considered in the BTE solution.
4-Please calculate the electronic band structures to also provide useful vision concerning the electronic nature of the studied system. This is also critical because only if the system is insulator or semiconductor the predicted value gives a comprehensive vision on the thermal transport otherwise for metallic systems the thermal conductivity can be affected considerably by electron phonon interactions.
5-Another possible way to accelerate the BTE calculations is to use machine learning interatomic potentials, as it has been shown recently for MTP+ShengBTE, generally MTPs could reproduce the phonon dispersion relation more accurately as compared with the current study.

Reviewer 2 Report

The authors have studied the thermal transport property of indium selenide, which has recently attracted much attention as a 2D post-transition-metal chalcogenide with promising physical properties. The thermal conductivity has been calculated through two approaches, including the Green-Kubo method applied on a classical trajectory obtained using molecular dynamics simulations with deep-learned interatomic potentials. The other approach incorporates the quantum-mechanical effects through density-functional theory for the calculation of the phonon dispersion, while using Boltzmann transport equation for computing of the thermal conductivity. The authors have found that the approach based on dynamics simulations results in a thermal conductivity comparable to the experimental values, while the static calculations excludes the possibility of four-phonon scattering processes and leads to an inaccurate value for the thermal conductivity.

Although the current work appears as a natural continuation of previous works of the authors in this field, the topic is timely and the manuscript is very well written and therefore, easy to follow. The presented results are supported by adequate discussions and the computational models have been justified and discussed well.

Therefore, I believe that the manuscript is of high interest to the community, both methodologically and application-wise, and can be published as it is.

Reviewer 3 Report

Authors of this work have investigated the lattice thermal conductivity of monolayer InSe using a DP model, and have claimed that they could reproduce the experimental results.

1-     Although the data provided were generated using an ML approach, authors should supply their ML code and datasets so readers of this paper can reproduce various plots reported in this ms. Otherwise, it is difficult to believe whether the results reported are trustable.

2-     The quality of English is not high. Authors should carefully read their paper to eliminate most of the typos and grammatical errors before resubmission. The corrections made should be highlighted in the revised ms.

3-     The phonon dispersion features need rigorous discussion, which is missing in the ms.

4-     What were the sizes of the supercells (in Å) used for phonon analysis?

Suggested to the authors for improvement. 

Round 2

Reviewer 1 Report

Authors have addressed my comments, I thus recommend the acceptance of this manuscript for the publication.

Reviewer 3 Report

Authors of this work have replied to all my concerns. I am OK with the revised version of the ms. It may now be recommended.

--------------